# Targeted volume imaging reveals early vascular interactions of Lyme disease pathogen in skin

Martin Strnad [1,2], Jiří Týč[1], František Kitzberger [1,2], Jana Kopecká[1], Ryan O. M. Rego [1,2] & Marie Vancová [1,2] ✉

Although the contours of the dissemination pathways of human pathogenic spirochetes in the vertebrate hosts are known, detailed high-resolution information on these processes remain lacking. In this study, we establish an efficient serial block-face scanning electron microscopy workflow incorporating semi-automatic AI-driven segmentation to investigate the architecture of early events following the deposition of *Borrelia burgdorferi* at the tick bite site in mice. We capture evidence of *Borrelia* penetrating the lymphatic endothelium via both transcellular and paracellular routes and observe its early presence within the lumen of the lymphatic vessel. The multistep process of transcellular migration is documented in detail, showing sequential invagination and encasement of shorter *Borrelia* segments by the lymphatic endothelial cells during intravasation. Our findings reveal that the first contact of *B. burgdorferi* and blood vessels is not random but involves close interactions with pericytes. We also capture the infiltration of immune cells in the skin and their interactions with invading bacteria. Altogether, these observations suggest that *Borrelia* strategically targets vascular regions with lower mechanical resistance to breach the endothelial barrier, thereby enhancing its dissemination.

Robust dissemination and barrier traversal are essential capabilities for pathogenic spirochetes, enabling them to invade and spread throughout the body of a host[1,2]. *Borrelia (Borreliella) burgdorferi* (*Bb*) is the causative agent of Lyme disease (LD) and is transmitted to humans by *Ixodes* spp. ticks[3]. To establish a disseminated infection, these spirochetes must overcome a multitude of physical obstacles, including entry into and exit from the blood and lymphatic circulatory systems[4,5]. The vascular barrier is primarily composed of endothelial cells (ECs) but is functionally supported and mechanically stabilized by a complex microenvironment that includes pericytes (PCs), basement membrane (BM), smooth muscle cells, fibroblasts, macrophages, and mast cells[6].

Despite advances in understanding the dissemination routes of *Bb*, much of the current knowledge is derived from in vitro and low-resolution in vivo microscopy studies focused on traversal of blood vasculature[5,7–11]. The literature is contentious, and it remains unclear whether vascular transmigration occurs primarily via transcellular[10,12–14] or paracellular pathways[5,9,13]. Notably, the absolute majority of these studies were designed in a way that they followed the process of extravasation−introducing *Bb* on the apical side of cultured ECs or into the lumen of the vessels and tracking its movement toward the basal side of ECs/perivascular space. A recent exception is a study that visualized intravasation using an advanced 3D microvessel model[9]. In that system, *Bb* was observed to cross the endothelium rapidly at cell-cell junctions. While 3D engineered vessels represent a significant step forward[15], they still fall short of fully replicating the mechanical strength and structural characteristics of native vascular tissue[16]. In vitro prepared EC monolayers lack perivascular support and

[1]Institute of Parasitology, Biology Centre, Czech Academy of Sciences, České Budějovice, Czechia. [2]Faculty of Science, University of South Bohemia, České Budějovice, Czechia. ✉e-mail: vancova@paru.cas.cz

a mature BM. As such, the barrier function of the endothelium is reduced in these models.

After transmission, *Bb* initially establishes itself in the skin at the site of the tick bite. *Bb* migrates with ease in the interstitial extracellular matrix (ECM)[17,18], causing a local infection often manifesting as erythema migrans in humans[19]. While translational motility is a key driver of infection[20], studies indicate that *Bb* adopts at least four distinct motility states in the skin[21]. These include a passive, non-motile state; a wriggling state where the bacterium undulates in place without moving forward; a lunging state where one end remains fixed while the rest of the body undulates and flexes; and a translocating state associated with directional migration[21]. The spirochetes do not remain locked in a single state. Instead, they switch between motility modes on a timescale of roughly 100 s, shifting from lunging to translocating or wriggling as needed. It has been hypothesized that the stationary states likely arise from bacterial adhesion to the ECM, where spirochetes remain motile and exert forces on the ECM but fail to translocate[21]. To date, the biological relevance of these stationary motility states has remained undetermined[22].

The increased use of volume electron microscopy (vEM) in pathogen research reflects a broader trend of incorporating advanced imaging techniques to study microbial ultrastructure and morphology[23]. Despite its growing popularity, high-resolution, large-volume structural insights into pathogen interactions within their native host environment remain largely unexplored[24]. Serial block-face scanning electron microscopy (SBF-SEM) offers substantial advantages in imaging larger tissue volumes at nanoscale resolution but often requires extensive time and computational resources[23]. To overcome these limitations, we employed an efficient sampling-based imaging strategy, where lower-resolution overview images were initially acquired to systematically and rapidly identify biologically significant regions. Subsequently, targeted regions were selectively imaged at higher resolutions, optimizing both imaging efficiency and speed. This interactive sampling strategy allowed us to capture rare yet biologically critical interactions in significantly reduced imaging times compared to conventional approaches, thus efficiently combining high resolution with large-volume coverage.

Using this approach, we followed the cascade of events at the cellular and subcellular level that arise after deposition of human infectious *Bb* into common tick bite areas in mice. We focused on (a) a detailed characterization of the transmigration pathways utilized by the pathogen to spread within the host, (b) ultrastructural analysis of borrelial cell envelope, with special emphasis on the possible shedding of outer membrane vesicles (OMVs), and (c) the interactions of *Bb* with immune and connective tissue cells. We demonstrate that *B. burgdorferi* rapidly engages with host tissue microenvironments and employs distinct strategies to initiate dissemination.

## Results

Injecting $4 \times 10^7$ spirochetes at the tick bite site is significantly higher than natural transmission levels, but it ensures an adequate sample size for this detailed imaging study. The definitive number of *Bb* transmitted during a tick bite can range from hundreds to thousands of spirochetes, depending on the tick's stage and *Bb* strain[25,26].

We imaged two areas: the ear and the dorsal skin. The total area examined for the ear measured $700 \times 500 \, \mu m$, with 700 slices of 60 nm thickness, covering a total volume of $0.015 \, mm^3$. The voxel size was $7 \times 7 \times 60 \, nm$. For the dorsal skin, the imaged area was approximately $1000 \times 450 \, \mu m$, with 850 slices of 80 nm thickness, covering a total volume of $0.027 \, mm^3$ and a voxel size of $7 \times 7 \times 80 \, nm$. Images at the specified resolution were acquired only from selected regions of interest (ROIs), which were determined based on large overview images acquired approximately every 20–30 sections, where potential ROIs were systematically searched for at each step. This imaging strategy allowed us to efficiently capture rare interactions in a greater

number between *Bb* and blood vessels or immune cells distributed across a large tissue volume in a reasonable timeframe, while maintaining high resolution and imaging speed. This interactive approach allowed us to refine our selection dynamically, adding more promising ROIs, continuing with previously identified relevant ones, and discontinuing acquisition in areas where no meaningful interactions were detected. This strategy is particularly valuable when conventional correlation techniques are not feasible and when objects of interest, such as relatively large *Bb* cells (which can reach lengths of up to $30 \, \mu m$), require targeted high-resolution imaging.

In the ear area, we observed epidermal and dermal cell layers, adipocytes, collagen fibers, lymphatic capillaries, blood vessels, immune cells, and fibroblasts (Supplementary Fig. 1a, b). In the dorsal skin, the tissue was located at the interface between the hypodermis and the muscle layer, where we identified blood capillaries, muscle bundles, immune cells, and collagen fibers (Supplementary Fig. 1c, d). Characteristic features used for the identification of key cell types are shown in Supplementary Fig. 1e–i. A more detailed description of the morphology of blood and lymphatic capillaries is provided in previous studies[27,28].

### Pericyte targeting by *Bb*

During dissemination, *Bb* must overcome several significant cellular barriers, with one of the most critical being the endothelial barrier of blood vessels. Intravasation is a rare event even under in vitro conditions, where 19 intravasation events could be tracked after inoculation of $1 \times 10^7$ spirochetes[9]. To capture the rare transmigration events, we specifically selected the North American species *B. burgdorferi* sensu stricto, which is known to disseminate early, in contrast to the major European species *B. afzelii*, which spreads more slowly and locally[29]. In total, we identified 18 areas where *Bb* was in close contact with the capillary surface (Figs. 1, 2 and 4, Supplementary Figs. 2–4). A "contact" refers to an area where *Bb* was located within 20 nm of the capillary surface[30]. Intriguingly, *Bb* appears to preferentially establish initial contact with PCs, which typically cover a smaller portion of the capillary surface. In regions where *Bb* interacted with blood capillaries, PC coverage ranged from 26% to 38%, averaging 28% across the capillaries, while the ECs accounted for the remaining 72% of the exposed capillary surface. Complete data on PC coverage percentages are presented in Table 1.

We analyzed ten *Bb* cells to quantify the percentage of *Bb* surface interacting with either ECs or PCs (Fig. 1). In four cases, *Bb* exclusively interacted with PCs, with the interacting regions predominantly located in the midsection of the bacteria (Fig. 1a–d, blue arrowheads). The

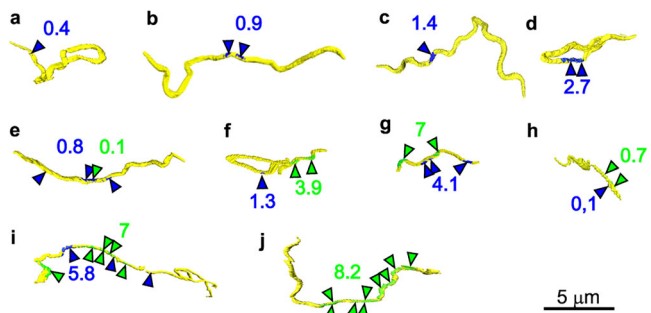

**Fig. 1 | Quantification and spatial distribution of *B. burgdorferi* contact areas with pericytes and endothelial cells in blood capillaries.** *Bb* spirochetes are shown in yellow, their contact zones with pericytes and endothelial cells in blue and green, respectively. Arrowheads indicate larger contact regions and their distribution. Values refer to the percentage of the *Bb* surface engaged in contact with each cell type (color-coded accordingly). Further details can be found for each *Bb* model: **a** (Supplementary Fig. 2a); **b** (Fig. 2a), **c** (Supplementary Fig. 3c), **d** (Supplementary Fig. 2b), **e** (Fig. 2b), **f** (Supplementary Fig. 4a–yellow *Bb*), **g** (Supplementary Fig. 4c), **h** (Supplementary Fig. 4a–orange *Bb*), **i** (Fig. 4), **j** (Fig. 3).

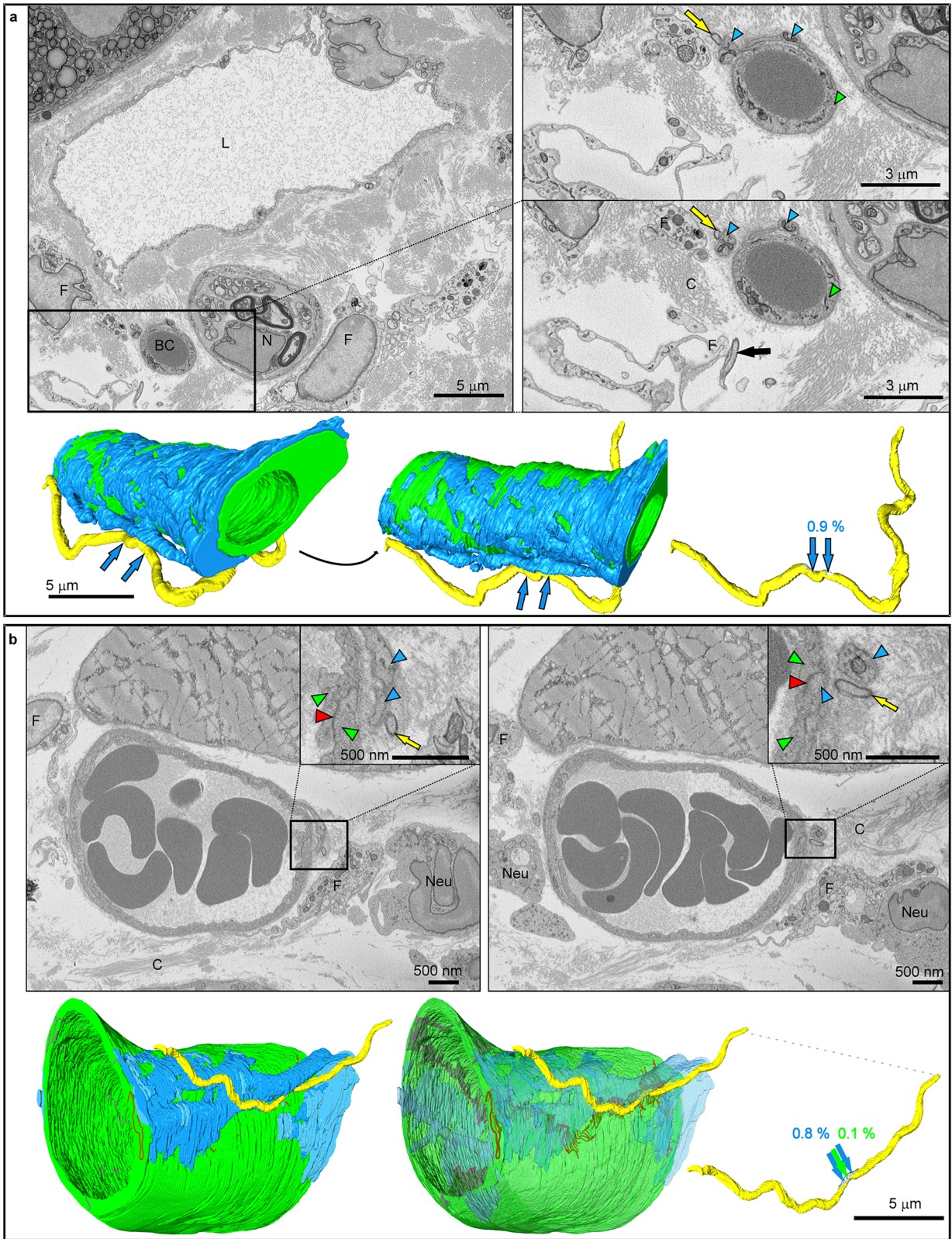

**Fig. 2 | Pericyte targeting by *B. burgdorferi*.** Representative images from SBF-SEM datasets are shown in grayscale, with corresponding 3D models. The percentages indicate the *Bb* surface area in contact with the pericyte (blue arrows) and endothelial cells (green arrows). **a** Blood capillary from the dermis of the ear. *Bb* in contact with the pericyte projection (yellow arrow) and the fibroblast (black arrow). **b** Capillary located in the hypodermis of the dorsal skin. *Bb* interacts with the pericyte protrusion, which covers the tight junction (red, red arrowheads) between endothelial cells. Structures are marked as follows: *Bb* (yellow, yellow arrows), blood capillary (BC), endothelial cell (green, green arrowheads), nerve (N), fibroblast (F), collagen (C), lumen of lymphatic capillary (L), pericyte projection (blue, blue arrowheads), neutrophils (Neu). For further details, see Supplementary Movies 1 and 2. Values refer to the percentage of the *Bb* surface engaged in contact with each cell type (color-coded accordingly).

**Table 1 | Quantitative information as to the distribution of pericytes and their contact with _B. burgdorferi_**

| Figure | Capillary PC coverage (%) | Borrelial surface in contact with (%) | |
|---|---|---|---|
| | | EC | PC |
| 2A | 38 | 0.00 | 0.93 |
| 2B | 21 | 0.11 | 0.75 |
| S2A | | 0.00 | 0.41 |
| S2B | NA | 0.00 | 2.73 |
| 3 | 26 | 8.18 | 0.00 |
| 4 | NA | 5.79 | 6.99 |
| S3C | NA | 0.00 | 1.40 |
| S4A (1) | NA | 3.98 | 1.34 |
| S4A (2) | NA | 0.70 | 0.10 |
| S4C | NA | 7.03 | 4.11 |

First column, pericyte (PC) coverage of blood microvessels. Second and third columns, surface area of _B. burgdorferi_ in direct contact with endothelial cells (ECs) and PCs, respectively, across 10 analyzed regions. Data were obtained using SBF-SEM.

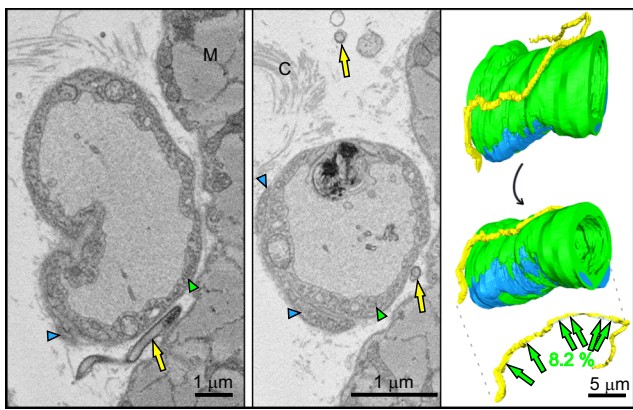

**Fig. 3 | Multisite adhesion of _B. burgdorferi_ to the blood capillary surface in dorsal skin.** Overview slices from two different depths of the SBF-SEM dataset and corresponding 3D model show _Bb_ (yellow, yellow arrows) adhering to the endothelial cells (green, green arrowheads). Pericytes are shown in blue or indicated by blue arrowheads. Muscle (M) and collagen (C). Value refers to the percentage of the _Bb_ surface engaged in contact with the endothelial cells.

percentage of _Bb_ surface in contact with PCs ranged from 0.4% to 2.7% in these instances (Fig. 1a–d).

Both Fig. 2 and Supplementary Fig. 2 provide a detailed depiction of _Bb_ adherence to PC projections. In the situation illustrated in Fig. 2a, the PC extends over the central region of the EC, away from junctional areas. _Bb_ contacts the tissue-facing side of the PC projection, using 0.9% of its surface. In contrast, Fig. 2b depicts a scenario where the PC projection closely follows and overlays the tight junction between adjacent ECs. _Bb_ touches the PC specifically at the site where the PC projection is notably thinned, and the junction beneath is partially exposed. Similar observations are also evident in Supplementary Fig. 2a, b. These figures also include 3D reconstructions that show the percentage of the _Bb_ surface involved in these interactions, as well as the specific regions of the bacterium in contact with PCs or ECs. The complete interplay of _Bb_ with blood capillaries is visualized in 3D across the full length of the bacterium in Supplementary Movies 1 and 2.

In five other cases, _Bb_ interacted with both PCs and ECs. PC-_Bb_ surface contact ranged from 0.1% to 5.8% and EC-_Bb_ contact ranged 0.1–7% (Table 1, Fig. 1e–i). In these instances, multiple distinct regions along the _Bb_ surface (marked by arrows in Fig. 1e–i) were in contact with the capillary surface.

Only a single _Bb_ was observed exclusively interacting with the EC, with the contact area comprising 8.2% of the bacterium's total surface, distributed over more than a half of the spirochete's length (Fig. 1j). This extensive surface contact between _Bb_ and the EC BM is shown in detail in Fig. 3. In this case, _Bb_ was wedged between muscle tissue and a capillary. Given the multiple contact points between _Bb_ and the capillary, as well as its positioning in such a confined space, we hypothesize that this interaction may represent a stationary phase of adhesion, allowing _Bb_ to evade phagocytosis by innate immune cells.

Figure 1i and in detail Fig. 4 show _Bb_ partially wedged between the PC and the EC. In total, this _Bb_ was in contact with 12.8% of its surface (Table 1), almost equally with the EC (7%) and the PC (5.8%). _Bb_ was observed to be invaginated into the PC projection, through which it completely penetrated into the ECM (black arrow in Fig. 4d). Further along, it is seen wrapped around the continuation of this projection (Supplementary Movie 3). Although _Bb_ disrupted the interface between the EC and the PC, it did not reach beyond the BM (Fig. 4c, d). The spatial arrangement was EC–BM–_Bb_–PC. In this region, we also identified an unusual round structure (highlighted in orange in Fig. 4c and Supplementary Movies 3 and 4) and cell debris (marked in violet in Fig. 4b and Supplementary Movies 3 and 4). Another _Bb_ positioned between the PC and EC is shown in Supplementary Fig. 4c. PC was

again not tightly associated with the BM at the PC-EC boundary, indicating a mechanical disruption of this interface.

**Trans- and paracellular traversal of lymphatic capillaries by _Bb_**
Although we did not observe any evidence of _Bb_ entering the blood vasculature early after infection, we did visualize how _Bb_ navigates transcellularly toward the lumen of lymphatic vessels through a multistep process (Fig. 5a, b, Supplementary Movies 5–10). Initially, _Bb_ is positioned along the capillary surface, partially burrowing into induced invaginations with smaller segments of its body, which eventually become encased by the EC (Fig. 5a, Supplementary Movies 8–10). Most of the _Bb_ cell remains extracellular at this stage. In the next phase, when a major part of _Bb_ becomes encased within the EC (Fig. 5b), the bacterium adopts a more perpendicular orientation toward the barrier and limits the contact between the extracellular part of its body and the capillary surface (Fig. 5b).

In paracellular traversal, we observed that the leading end of _Bb_ was positioned between adjacent ECs (Fig. 5c, Supplementary Movies 11 and 12). In this region, the junction between the cells was loosened (Fig. 5c, pink arrow), while the EC processes above and below this point were tightly attached (Fig. 5c, white arrows). Although we did not observe the instance of _Bb_ entering the capillary lumen, where _Bb_ would be positioned partially inside the lumen and partially within the endothelium, we found two spirochetes inside the lumen (Supplementary Fig. 5, Supplementary Movie 13). This indicates that penetration into lymphatic capillaries takes place shortly after deposition of _Bb_ into the skin, in contrast to its entry into blood capillaries.

**Interactions with noncellular ECM barriers—_Bb_ located inside the collagen bundles**
Earlier studies have shown in vitro that the spirochete can degrade components of the ECM, and this ability is directly connected to its invasiveness and ability to spread within the host[31–33]. In this study, we observed the pathogen in its natural host environment to evaluate its tissue-degrading activity during dissemination in vivo. Our data suggest that _Bb_ does not rely heavily on proteolytic activity during the initial stages of dissemination. We frequently observed _Bb_ localized within collagen-rich regions of the interstitial ECM, where _Bb_ aligned itself in the same orientation as collagen bundles (Fig. 6, Supplementary Movie 14). The movement of _Bb_ within the collagen bundles consistently followed the orientation of the fibers. This was evident in both transverse and longitudinal cross-sections of the collagen bundles. In transverse sections, _Bb_ appeared as cross-sections of the spirochetes,

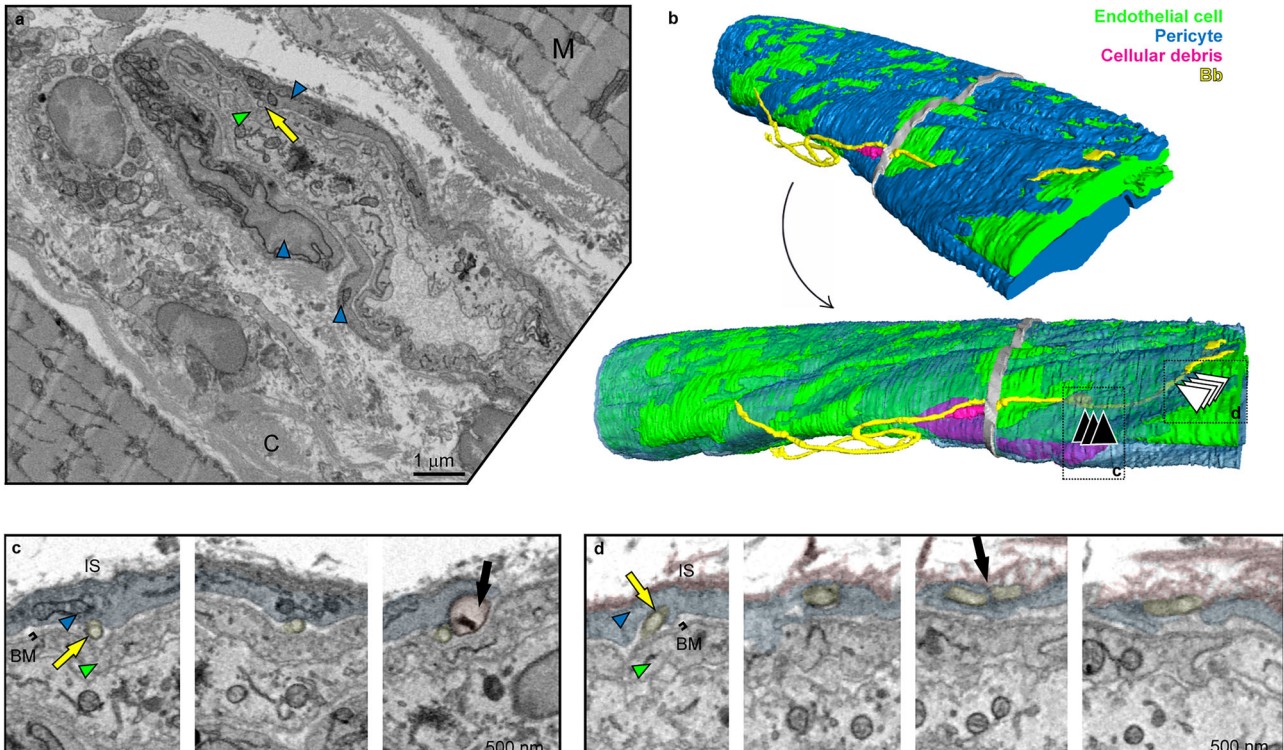

**Fig. 4 | Pericyte-penetrating *B. burgdorferi* partially wedged between a pericyte and an endothelial cell in the dorsal skin. a** Overview representative slice from the SBF-SEM dataset showing a cross-section of *Bb* (yellow arrows) located between the pericyte (blue arrowheads) and the endothelial cell (green arrowheads). **b** Two 3D model views of the same capillary from different angles show two adhered *Bb* cells. In the bottom model, pericytes are rendered transparent to expose the localization of one *Bb* beneath the pericyte, along with cellular debris located above (magenta) and beneath (violet) the pericyte. Missing sections are shown in gray.

**c** Representative slices from the region of the dataset indicated by black arrowheads in (**b**) reveal an unidentified structure adjacent to the *Bb* (black arrow). **d** Slices from the area marked in (**b**) by white arrowheads show *Bb* invaginated into the pericyte protrusion at the putative entry site, and penetration through its membrane at the putative exit site (black arrow). For color coding, refer to (**a**). Basement membrane (BM), interstitial space (IS). Additional details are provided in Supplementary Movies 3 and 4.

while in longitudinal sections, *Bb* shifted its appearance in accordance with the fiber orientation. This suggests that migration of *Bb* is not random, but highly coordinated with the structural organization of the ECM. *Bb* could reach the capillary surface regardless of the presence (Figs. 2a, b and 3) or absence (e.g., Supplementary Fig. 3a, b, e) of the collagen bundles near the vasculature, suggesting that these structures do not play a major role in the *Bb* intravasation process.

### Atypical morphologies of *Bb* and rare shedding of OMVs

In this study, we observed a very rare production of OMVs. A few OMVs-secreting *Bb* were observed within the interstitial ECM (black arrows in Supplementary Fig. 3b, c). In one case, *Bb* located within a deep invagination of a lymphatic EC secreted OMVs (black arrows in Fig. 5a). Surprisingly, in two instances near blood capillaries, *Bb* displayed atypical morphologies. In one, the *Bb* cell appeared as a mid-folded cylinder enclosed by an outer membrane (Supplementary Fig. 4b; both parts of one *Bb* are marked in yellow and orange in the 3D model). The other atypical structure was observed in Supplementary Fig. 4a (marked in purple in cross-section and in the 3D model), which belongs to a *Bb* cell highlighted in orange, while the other *Bb* located in the same area is in yellow. It appears that the structure marked with a black arrow shown in Fig. 4c, Supplementary Movies 3 and 4, is also of borrelial origin.

### Cellular infiltration at the tick bite site

Under normal, non-inflamed conditions, the ECM contains few immune cells, as most reside in lymphoid organs, blood, or specific immune-privileged tissues[34]. Nevertheless, we found mast cells in both

tissue types, and in their surroundings were free granules (presumed degranulated mast cells, Fig. 7a, asterisks) as well as neutrophilic granulocytes, which typically possess the nucleus with several lobes connected by a thin filament. The migration of immune cells from the blood vasculature is captured in Fig. 7b (arrows) and Supplementary Movie 15. In several neutrophils, we imaged a phagosome with one phagocytosed *Bb* (Fig. 7c–e, Supplementary Movie 16). In Fig. 7f, g, Supplementary Movie 17, neutrophils had two phagosomes plus another *Bb*, located in the filopodium of the neutrophil, with its other end outside the neutrophil.

## Discussion

The great challenge for biological microscopists is the interpretation of complex structures and interactions from two-dimensional and low-resolution images. To define the spatio-temporal features of microbial infections, structural data need to be obtained in the natural host environment in three dimensions with high resolution. Due to the wave-like morphology and typically low abundance of pathogenic spirochetes in tissues, it is challenging to accurately identify the bacteria from random 2D electron microscopy projections. To reliably confirm the presence of *Bb*, one must either employ a correlative approach[35] or conduct a complete 3D reconstruction of the spirochete. Determining the critical interactions within the host environment will help identify key mechanisms of pathogenesis and uncover new therapeutic targets[36].

In this study, we employed SBF-SEM to visualize early interactions of *Bb* with host tissues. SBF-SEM enables imaging of relatively large tissue volumes at high ultrastructural resolution[23], striking an optimal

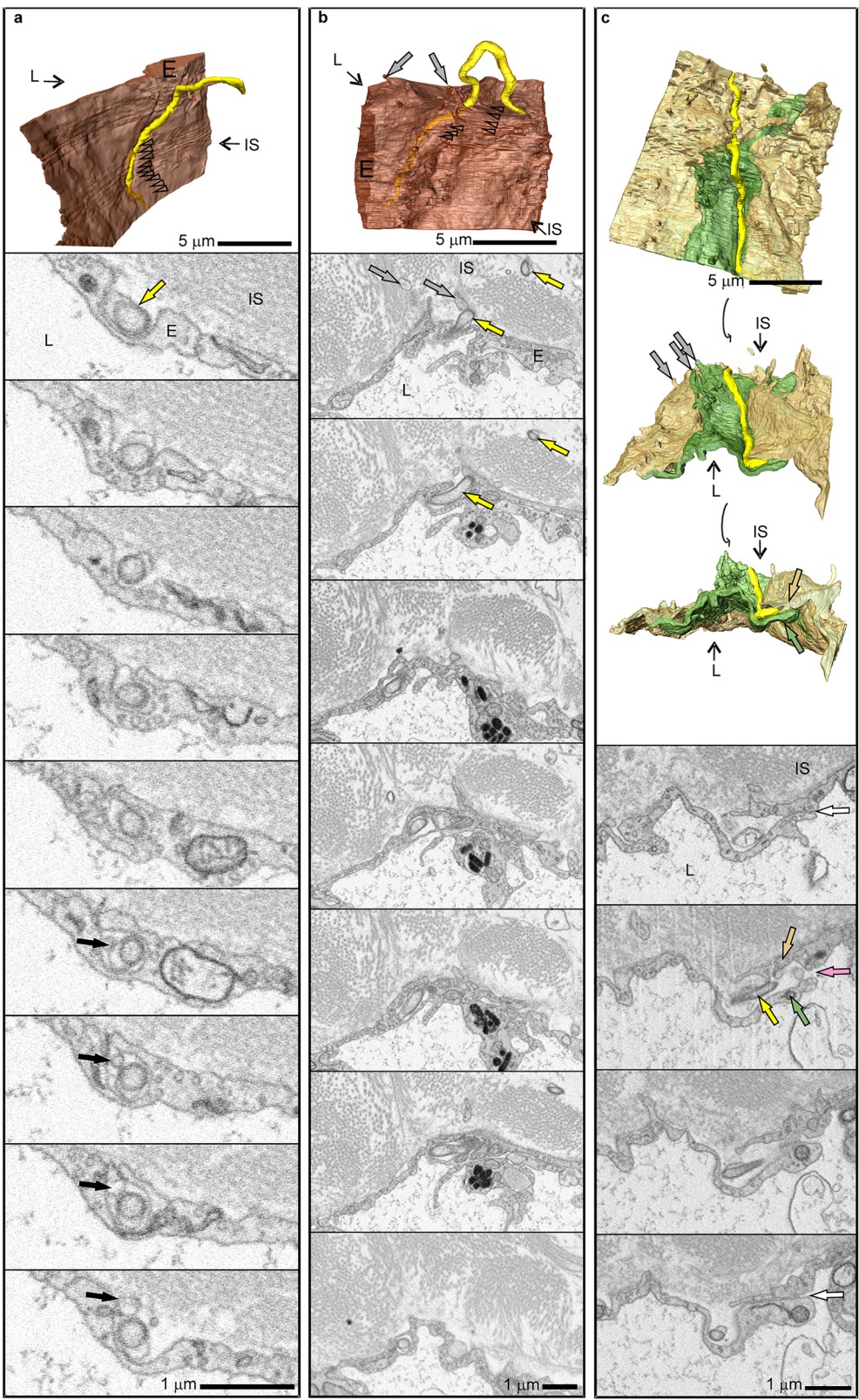

**Fig. 5 | Transcellular and paracellular traversal of lymphatic capillaries by *B. burgdorferi*.** The 3D models at the top are accompanied by representative sections from SBF-SEM datasets, arranged in columns. Transparent arrowheads in the 3D models point to the regions shown in the micrographs. **a**, **b** Transcellular migration through endothelial cells (E, brown), facilitated by sequential embedding of *Bb* in the endothelial cell. **a** Leading end of *Bb* is stably embedded in the endothelial cell, while the next segment of *Bb* is being enclosed (arrowheads). Black arrows indicate an outer membrane vesicle. **b** Major invasion of an endothelial cell, while the trailing end remains free to move. **c** Paracellular traversal. *Bb* located in a space between neighboring but locally disconnected (pink arrow) endothelial cells (light brown and green arrows). The cells were tightly apposed in the upper and lower sections of the dataset (white arrows). Protrusions of endothelial cells into the interstitial space (IS) are shown by gray arrows. Lumen (L). *Bb* (yellow in the 3D models, yellow arrows). For further details, see Supplementary Movies 5–12.

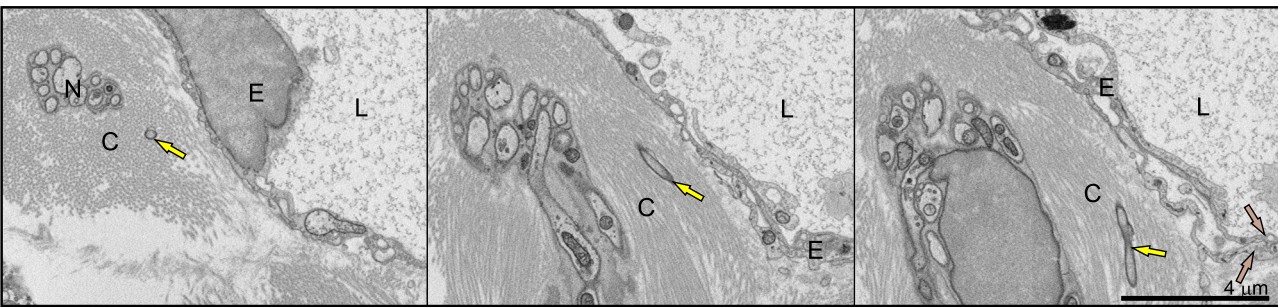

**Fig. 6 | Visualization of *B. burgdorferi* in the ear positioned along collagen fibers, following their orientation.** In the transverse cross-section (left) of collagen bundles, a transverse cross-section of *Bb* is also visible. In tangential (middle) and longitudinal sections (right), the appearance of *Bb* shifts accordingly.

Endothelial cell (E), nerve (N), collagen (C), lumen of the lymphatic capillary (L), *Bb* (yellow arrows), overlapping ECs (brown arrows). For further details, see Supplementary Movie 14.

balance between sample volume, achievable resolution, and practical imaging speed. Although vEM encompasses several other modalities, such as FIB-SEM, which provides superior axial resolution due to thinner slicing, and array tomography, which preserves sections for subsequent re-imaging and molecular labeling, each has specific limitations[37]. FIB-SEM imaging is typically limited by significantly smaller imaging volumes and longer acquisition times, whereas array tomography requires labor-intensive handling and introduces challenges for larger sample volumes[38]. A traditional serial-section TEM approach, though offering excellent lateral resolution, remains highly labor-intensive and restricted to very small tissue volumes, making it unsuitable for studying the rare and dispersed interactions of pathogens within tissues. To overcome these constraints, we introduced an optimized imaging strategy for SBF-SEM employing initial sampling at lower resolution to rapidly identify promising ROIs, which were subsequently imaged selectively at higher resolution. This interactive targeted imaging approach substantially increased throughput and shortened acquisition times while preserving critical ultrastructural details required to accurately characterize rare events such as borrelial traversal of endothelial barriers.

Not surprisingly, we often found *Bb* to be present within collagen bundles (Fig. 6, Supplementary Fig. 3b, Movie 14). *Bb* is known to invade native collagen matrices[39] and also targets the collagen fibers during infection[40]. Collagen-binding activity of *Bb*, mediated by adhesins such as BBA33, plays a critical role in the early stages of mouse infection[41]. While *Bb* appears to degrade elastic and collagen fibers in later disease stages[40], there is no direct evidence of structural degradation during early dissemination in vivo. On the contrary, in a collagen-rich environment, the distribution of motion reversal events is significantly elevated, which indicates that *Bb* tends to alter its path rather than attempting to degrade and force its way through sturdy structures[9,42]. Although *Bb* produces outer membrane proteases[43], the quantity and diversity of these proteases are substantially limited compared to other invasive bacteria that use a wide range of exoenzymes such as elastases, collagenases, and hyaluronidases to degrade ECM components[44–46]. In line with this, we did not observe any visible damage to the collagen fibers within the interstitial ECM. *Bb* seems to navigate around these internal obstacles, which provide natural pathways for migration through connective tissue.

The behavior of *Bb* may reflect a broader evolutionary strategy to balance between efficient tissue invasion and immune evasion. By avoiding overt destruction of host tissues, *Bb* may reduce the likelihood of triggering host defense mechanisms that could thwart its spread[47]. The lack of the OMVs secreted from the bacterial surface, as observed in this study, further supports this strategy. OMVs are often produced in response to stressful conditions such as exposure to antibiotics, oxidative stress, or immune system attack[48]. Shedding OMVs in *Bb* can be observed in culture[49]. OMVs from *Bb* were shown to

be on average 33 nm in diameter and contained immunogenic proteins such as OspA, OspC, p39, and peptidoglycan[49]. Our observations contrast sharply with the significant shedding of OMVs in the tick midgut during the early feeding of *Bb*-infected ticks[50]. This suggests that *Bb* may actively remodel its membrane in preparation for transmission, discarding obsolete or tick-specific proteins. Shedding these components early, while still in the tick, could help the spirochete avoid triggering an unwanted immune response upon entering the host.

Pathogenic spirochetes rely on effective mobility and barrier crossing to infiltrate the host. The in vitro transendothelial migration studies based on cellular monolayers have largely been limited to *Bb*–EC interactions[9,10,12,13], neglecting the complexity added by other components of the microvessel wall, specifically PCs and the shared condensed BM. The three key structural elements—ECs, PCs, and BM—work together as a coordinated system that regulates vessel integrity and function[51]. In addition, the endothelial glycocalyx, which reaches up to 7 μm in the blood vessels, is attenuated under the static conditions of in vitro cell culture[52]. Smooth muscle cells are involved in maintaining and regulating the endothelium's integrity and function[51]. Fibroblasts, present in the surrounding connective tissue, produce ECM components, which support the structure of the endothelium[53]. Mast cells are often found near blood vessels and play a role in inflammatory responses[54]. Also, leukocytes and platelets are vital for proper endothelial function by contributing to immune surveillance, tissue repair, and the regulation of vascular integrity[55].

In this study, we present the first evidence that *Bb* frequently targets PCs. Using SBF-SEM, we visualized the entire surface of *Bb* at high resolution and found that when *Bb* contacted the capillary surface with a small area, typically localized to one region, *Bb* interacted specifically with projections of PCs (Figs. 1 and 2, Supplementary Figs. 2 and 3c, Supplementary Movies 1 and 2). As the number of interacting areas along the *Bb* body increased, *Bb* also engaged with the ECs (Figs. 3 and 4, Supplementary Figs. 3 and 4). This recognition suggests a two-phase model of *Bb* vascular transmigration (Fig. 8). The initial phase likely represents a "probing" stage during which the bacteria survey the microenvironment. At this stage, *Bb* interacts with the vasculature using a smaller area of its cell body, without penetrating either the PCs, ECs, or the surrounding BM (Fig. 2). This is followed by a second phase characterized by a gradual increase in the number of endothelial contact sites, suggesting a transition from transient adhesion to a more stable interaction prior to traversal. Our observations also show that *Bb* tends to position itself between the ECs and the PCs (Fig. 4, Supplementary Fig. 4c, Supplementary Movies 3 and 4). Since PCs are known to mechanically stabilize the endothelium and influence vascular permeability[51], targeting PCs by pathogenic bacteria may compromise the barrier function of the endothelium and dampen the pro-inflammatory responses of ECs[56].

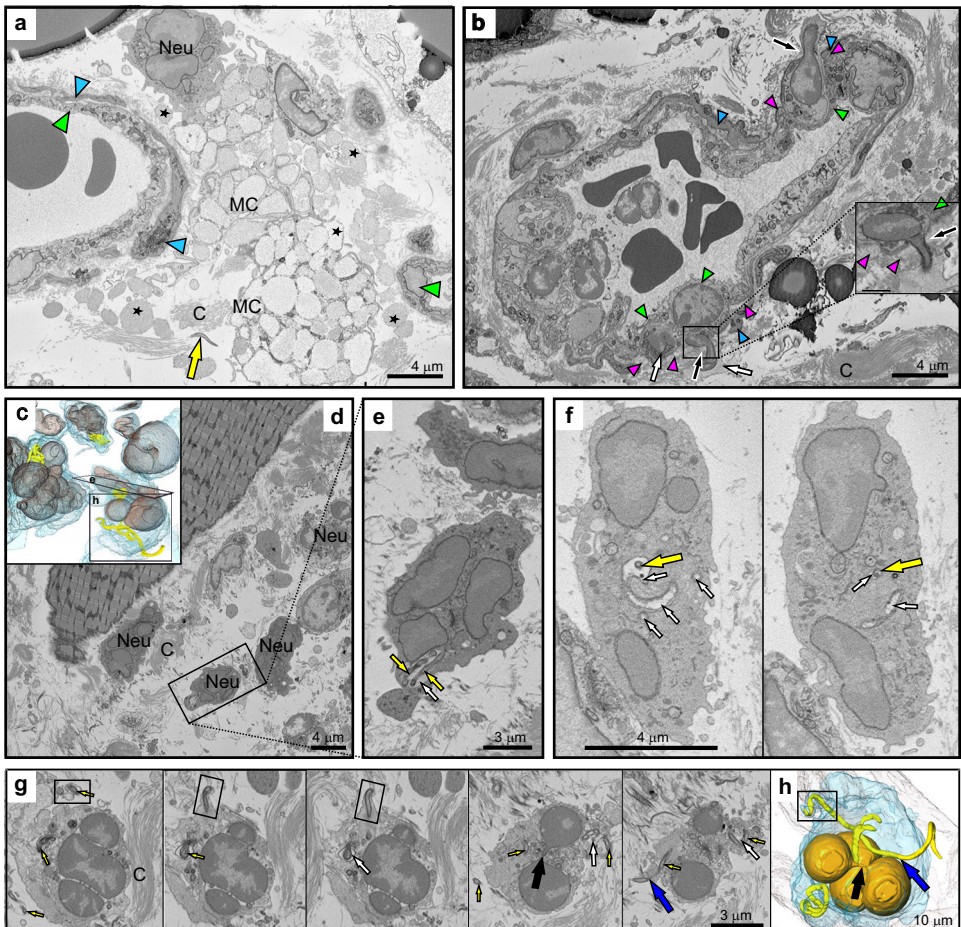

**Fig. 7 | *B. burgdorferi* interactions with immune cells. a** Infiltration of the interstitial space with mast cells (MC; granules released from MC are indicated by asterisks) and neutrophils (Neu). **b** Immune cells actively migrating from the vein into the interstitial space. Black arrows indicate areas where neutrophils traverse the basement membrane, which represents the narrowest area through which the immune cells pass. In the inset, this region is shown in a different plane. For further details, see Supplementary Movie 15. **c**–**f** 3D models and images from the SBF-SEM dataset showing *Bb* engulfed by neutrophils. Entire *Bb* is located in a spacious phagosome (**e**), while *Bb* in (**f**) is in a phagosome that coils. Phagosome (white

arrows). **g**, **h** 3D model and slices from the dataset showing a neutrophil containing two fully engulfed *Bb* and one *Bb* possibly escaping or in the process of being engulfed. This *Bb* is located within a tiny filopodium (black box) and crosses the neutrophil cell wall (blue arrow). Black arrows indicate tapering protrusions between nuclear segments (orange in 3D model). *Bb* (yellow arrows), pericytes (blue arrowheads), endothelial cells (green arrowheads), basement membrane (magenta arrowhead), collagen (C), phagosome (white arrows). **a** Ear skin, **b**–**h** dorsal skin. For further details, see Supplementary Movies 16 and 17.

This disruption may facilitate the ability of *Bb* to overcome this anatomical obstacle.

Despite the important role of PCs in vascular function, there is a lack of information about their targeting by pathogenic bacteria. Several viruses are known to infect, and some even replicate inside the PCs[57–62]. Investigating the mechanisms by which pathogenic bacteria target PCs is crucial for identifying the key molecules involved in cellular barrier transmigration. PCs exhibit a strong affinity for ECM components, particularly fibronectin, which is concentrated at the PC-EC interface[63,64]. PCs strongly prefer fibronectin over laminin for adhesion formation[63]. *Bb* expresses several fibronectin-binding proteins that facilitate its adhesion to host tissues[65], such as BBK32[66], RevA[67], RevB[67], and CRASP-1[68]. Consequently, fibronectin may be the crucial factor driving the contact and colocalization of PCs and *Bb*.

Altering the biomechanical stability of the endothelium is a recognized strategy employed by pathogenic organisms, including bacteria[69]. *Bb* can influence the stability and intercellular forces of the endothelium as part of their infection strategy[70]. PCs are located on the abluminal surface of capillaries and venules, wrapping around the ECs. Postcapillary venules and capillaries serve as key microvascular sites

for the migration of *Bb* into and out of the circulatory system[5]. In postcapillary venules, the PCs coverage is typically around 20–30% of the endothelial surface, but varies depending on specific studies and tissue types[51,71]. This coverage is generally lower than in capillaries. The lower PC coverage in postcapillary venules facilitates greater vascular permeability and immune cell trafficking[51]. In our observations, PCs covered an average of 28% of the vessel surface, based on measurements across four areas (Table 1).

LD spirochetes are thought to utilize the lymphatic system as an alternate means to disseminate to more distant tissues[4,72]. Notably, lymph nodes are often *Bb* culture positive sooner than other tissues near the inoculation site, indicating the involvement of the lymphatic capillaries for the early dissemination of *Bb*[73,74]. This aligns with our findings, as *Bb* was visualized in the lumen of a lymphatic capillary and in multiple areas attempting to enter lymphatic capillaries (Fig. 5, Supplementary Fig. 5, Supplementary Movies 5–12). Our observations indicate that during transcellular traversal of the endothelium, *Bb* does not initially adopt a head-first perpendicular orientation towards the barrier. Instead, spirochetes initiate transmigration in lateral alignment to the endothelium, via sequential invagination of shorter segments of the borrelial cell (Fig. 5a, b). *Bb* likely induces these invaginations

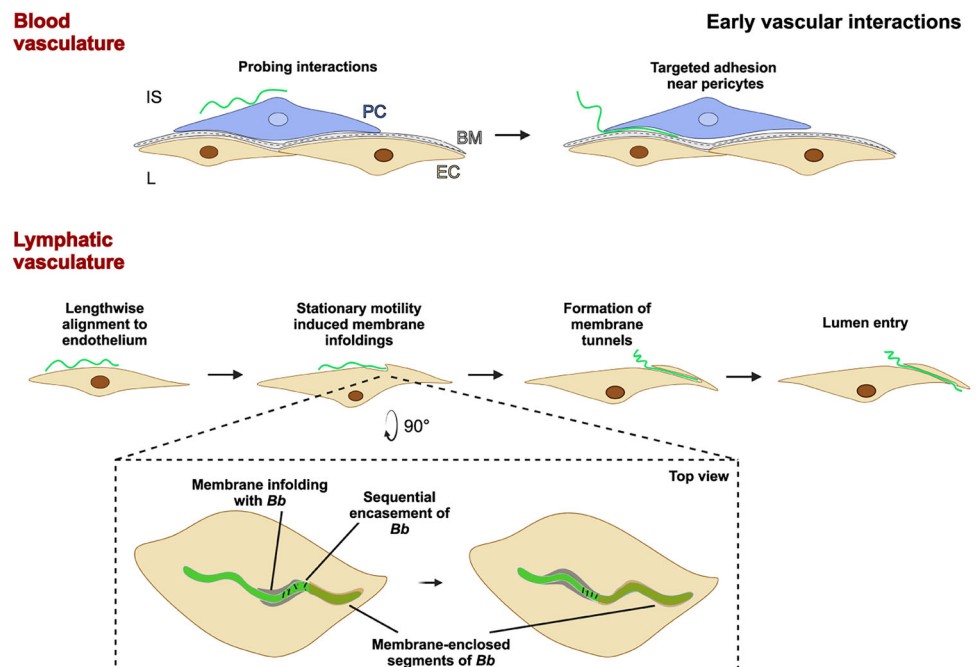

**Fig. 8 | Schematic illustration of early vascular interactions of *B. burgdorferi* and the proposed mechanics of transcellular intravasation.** Intravasation represents a critical step in systemic dissemination of *Bb*. The preparation process preceding the traversal of the blood endothelium includes at least two stages. The first phase seemingly involves probing, low-surface-contact interactions with the vasculature, primarily targeting PCs. In the second phase, *Bb* establishes multiple contact points with both ECs and PCs, transitioning from an exploratory stage to stable adhesion. *Bb* tends to localize between ECs and PCs, likely disrupting their function in maintaining vascular integrity. The BM appears to be the primary ana-tomical barrier to *Bb* intravasation, as no instances of *Bb* breaching this barrier were observed in this study. Regardless of the approach angle, *Bb* ultimately aligns itself lengthwise with the lymphatic endothelium before initiating traversal of the barrier. In transcellular traversal, *Bb* is sequentially encased in membrane infoldings of ECs. The stationary, motile, and non-motile states of *Bb*[21] likely play a key role in the formation of the membrane infoldings and embedding of *Bb* in the EC. In all stages, *Bb* remained separated from the cellular cytosol by the plasma membrane and reached the lumen of the lymphatic vessel enclosed in this tunnel-like structure. Interstitial space (IS), vascular lumen (L), basement membrane (BM), endothelial cell (EC), pericyte (PC). Created in BioRender. Strnad, M. (2025) https://BioRender.com/0b7pn5r.

mechanically, as after adhering to the microvessel, this bacterium often exhibits a back-and-forth motion along the surface[5,9]. As the bacterium progresses to the next stage and becomes mostly embed-ded in the cell (Fig. 5b), it repositions itself, aligning more perpendi-cularly with the barrier. During paracellular traversal, *Bb* exploits the loosened junctions of lymphatic ECs (Fig. 5c). Interestingly, *Bb* has once again been observed immersed in EC membrane infoldings, underscoring the importance of *Bb*–EC interactions in endothelial traversal. The overlapping processes of neighboring ECs are clearly detached ahead of the borrelial leading end (Fig. 5c, pink arrow), while those above and below remain tightly attached (Fig. 5c, white arrows). It is unclear whether *Bb* actively induces these openings or simply exploits pre-existing weaknesses in junctional integrity. What is evi-dent, however, given its extensive contact with the ECs, is that *Bb* moves forward by pushing against the surrounding structures rather than boring through them with its leading end.

A striking feature of *Bb* intravasation observed in this study was the lengthwise alignment of *Bb* along the vasculature, similar to the orientation during the initial stage of extravasation[5,75]. However, dur-ing extravasation, *Bb* is subjected to significant shear forces from blood flow, necessitating a firm grip on the vessel wall to prevent displacement. The lateral alignment observed in the static perivascular environment during intravasation is much less intuitive. Intravital microscopy studies of extravasation have indicated that before crossing the vessel wall, *Bb* remains anchored at a single point, with one end fixed while the rest of the body remains mobile[5,75]. Here, we demonstrate that a similar sequence of events occurred during the intravasation of lymphatic capillaries (Figs. 5 and 8). We observed that the anchorage site is stabilized not only by adhesive interactions but

primarily through the physical embedding of a part of the bacterial cell. Sequential encasement of shorter *Bb* segments by lymphatic ECs during intravasation also sheds light on the functional roles of the four known motility states of *Bb*. The translational motility state facilitates locomotion, while the wriggling and lunging states likely induce membrane remodeling by pushing the crests of the *Bb* wave onto the EC, leading to the formation of membrane infoldings. *Bb* then enters these membrane folds and exploits them en route to the vessel lumen. Finally, by remaining in a passive non-motile state, *Bb* likely avoids disrupting the encasement process, facilitating the seamless closure of the invagination. The 3D models confirmed the presence of fine pro-jections extending from the EC surface (Fig. 5), enwrapping the spirochetes.

Despite forming a formidable obstacle, the BM is often over-looked in studies of endothelial barrier traversal, which tend to focus on the EC lining as the main physical barrier[76]. The BM provides essential structural and mechanical support to the endothelium, and the stark structural differences between lymphatic and blood capil-laries seem to be a major factor enabling *Bb* intravasation. Lymphatic capillaries lack PCs and frequently have a discontinuous or absent BM[77,78]. In this study, the BM of lymphatic vessels was not detectable (Fig. 5), while the BM of blood capillaries was clearly defined (Fig. 4c, d). BM is rich in collagen IV, which typically self-assembles into a sheet-like network stabilized by a high degree of crosslinking and covalent disulfide bonds[16]. Electron microscopy studies have even shown that leukocytes easily traverse the thick layer of ECs that line venules but stall when they reach the BM. Conceivably, the BM might represent a more significant barrier to transmigration than the endothelium itself[79,80].

Our observations indicate that *Bb* struggles to cross an intact BM during intravasation of blood capillaries (Figs. 2 and 4). While no spirochetes were detected beyond this barrier in blood vessels, one was seen inside the PCs (Fig. 4), implying that the plasma membrane penetration might be less challenging for *Bb* than traversing the BM. These findings align with those of Guo et al.[9], who demonstrated that the crossing of the ECs takes <4 s, yet *Bb* often remains tethered to the endothelium for up to 100 s prior to intravasation. That period is likely used to breach the base layer of the endothelium. However, given the incompletely developed BM in the vascular models, the observed time ratio (~25:1) between *Bb* wriggling along the endothelium and actual transmigration likely does not accurately mirror the dynamics of intravasation in vivo.

Pathogens typically cross the BM using two main strategies[76]. The first involves enzymatic degradation, where microbial or host-derived proteases break down BM components. The second is cell-mediated traversal, in which pathogens exploit immune cell trafficking to ferry them across the BM. *Bb* seems to take a distinct approach to navigate ECM-based structures. In the interstitial space, it utilizes its large arsenal of surface adhesins and powerful periplasmic flagella, which generate substantial propulsion forces[81], to maneuver through the barrier[18,82]. The average pore size of approximately 50 nm in the collagen–laminin network of the BM suggests that only very small particles can pass through this barrier[76,83]. Given that *Borrelia* spirochetes have a diameter of 200–300 nm, they are too large to migrate effortlessly across the BM. Interestingly, the BM tends to be thinner in regions where PCs are closely linked to ECs, compared to areas where it only supports ECs[84,85]. These modifications may create a more permissive region that *Bb* can exploit to gain entry into the bloodstream.

A hallmark of tick engorgement in vertebrates is the infiltration of host immune cells on or around the tick bite site, which does not interfere with the blood feeding process[86]. Although the timing of cellular recruitment and the composition of the infiltrates can vary to some degree from species to species, all hosts exhibit intense cellular infiltration at the bite site within the first 6 h of tick attachment. Studies in laboratory mice have shown that these infiltrates include a mixed population of inflammatory cells, such as neutrophils, lymphocytes, eosinophils, and basophils, along with occasional tissue macrophages. This cellular response is accompanied by clear signs of vascular injury in the adjacent dermis, such as vascular dilation and erythrocyte extravasation[87]. Also, mast cells play a critical role in this response[88]. *Bb* induces mast cell activation. Once activated, mast cells trigger a series of inflammatory responses, including the activation of immune cells and the release of pro-inflammatory cytokines. Using SBF-SEM, we visualized the extent of tissue infiltration, particularly by neutrophils and mast cells (Fig. 7a, d). Neutrophils, especially when present in higher densities in localized areas, were capable of *Bb* phagocytosis (Fig. 7c–g, Supplementary Movies 16 and 17). Ultrastructural analysis revealed no structural alteration, suggesting the direct effects of mast cells on *Bb* (Fig. 7a). The immune response, while targeting the infection, may also compromise the integrity of the endothelial barrier. This weakening could facilitate the entry of *Bb* into the capillary lumen, similar to how it promotes *Bb* extravasation[14].

To summarize, a targeted, high-resolution SBF-SEM imaging approach allowed for efficient identification and visualization of rare but biologically important events of *Bb* infection in large tissue volumes. At the tick bite site, we observed that *Bb* often migrated through the interstitial ECM along collagen fibers, and shedding of OMVs was limited. Immune cell infiltration was evident, particularly by neutrophils and mast cells, with neutrophils actively phagocytosing *Bb*. The study reveals that shortly after infection, *Bb* already interacts with the blood microvascular environment, predominantly targeting PCs, but does not yet breach the lining of ECs or their BM. In contrast, we detailed how *Bb* penetrates the lymphatic endothelial barrier,

confirming both transcellular and paracellular intravasation. The discontinuous nature of lymphatic vessels, including the poorly defined BM, makes them a more surmountable barrier than blood vessels and a preferential target for early systemic spread. Due to its limited repertoire of outer membrane proteases, *Bb* facilitates its spread by either exploiting weak anatomical barriers or weakening their structural integrity by disrupting the function of key stabilizing cells. The motile but stationary states of *Bb*, previously not assigned a specific function, have been linked to inducing membrane deformations and infoldings in ECs. The undulation of the *Bb* body apparently not only drives translational motion but also exerts significant forces perpendicular to the direction of wave propagation, thus facilitating membrane remodeling. By studying critical host-pathogen interactions in 3D at high resolution within the native tissue context, we can gain a much deeper understanding of the mechanisms driving pathogenesis.

## Methods

All animals used for the experiments were treated in accordance with the Animal Protection Law of the Czech Republic, no. 246/1992 Sb., ethics approval no. 161/2011. The animal experimental protocol was approved by the Czech Academy of Sciences Animal Care and Use Committee (8/2023/P).

### Bacterial culture

Bb914, a virulent derivative of the *B. burgdorferi* strain 297 expressing GFP, was used in this study[50]. *Borrelia* cultures were propagated at 34 °C in BSK II culture medium supplemented with 6% (vol/vol) rabbit serum until the culture reached mid-log exponential phase.

### Mice infection

Uninfected nymphal *Ixodes ricinus* ticks ($n = 5$; obtained from the tick-rearing facility, Institute of Parasitology, Biology Centre, Czech Academy of Sciences; maintained at 24 °C, 95% humidity, and a 15/9 h light/dark cycle) were allowed to feed for 2 days on 8-week-old female C3H/HeN mice (bred in the institute's animal facility) at two sites: the ear and the back. The mice were housed in plastic cages with wood-chip bedding with a constant temperature of 22 °C, a relative humidity of 65%, and under a 12 h ON/12 h OFF light cycle. The bacteria were enumerated using a Petroff-Hausser counting chamber, and the injection volume was adjusted to 50 μL. Bb914 was injected subcutaneously into the tick bite site ($4 \times 10^7$ spirochetes). Two hours after inoculation, the mice were euthanized, and skin tissue samples were dissected.

### Processing for EM

These samples were immediately fixed in 2.5% glutaraldehyde, 2% formaldehyde, and 2 mM calcium chloride in 0.15 M cacodylate buffer for 2 h at room temperature, followed by overnight fixation at 4 °C. After fixation, the samples were rinsed in cacodylate buffer and post-fixed in 2% osmium tetroxide ($OsO_4$) for 1.5 h at room temperature (RT). They were then treated with 2.5% potassium ferrocyanide for 1.5 h at RT. The tissues were washed three times in distilled water ($dH_2O$), followed by incubation in 1% thiocarbohydrazide for 2 h. After rinsing in water, the samples were postfixed again in 2% $OsO_4$ for 4 h at RT. The samples were washed in water and immersed in 1% aqueous uranyl acetate overnight at 4 °C, followed by another wash in water. Walton's lead aspartate solution was prepared as follows: 0.998 g of L-aspartic acid was dissolved in 250 mL of $ddH_2O$ to prepare a 0.03 M stock solution. To this, 0.066 g of lead nitrate was added to 10 mL of the L-aspartic acid solution, adjusting the pH to 5.5 with 1 N KOH. The samples were stained for 2 h at 60 °C in Walton's lead aspartate solution, then washed in water. The samples were then dehydrated in a graded acetone series (30%, 50%, 70%, 90%, 100%) for 2 × 7 min at 4 °C at each step. The samples were infiltrated with 25%, 50%, and 75% Hard Plus Resin 812 (EMS, 14115) in 100% acetone for 1 h at RT at each step,

with agitation. Following this, the samples were infiltrated with fresh 100% Hard Plus Resin 812 for 6 h at RT. The resin was then polymerized for 48 h at 60 °C.

After polymerization, the resin blocks were trimmed, and the tissue was exposed on all sides. The blocks were then mounted onto aluminum specimen pins using cyanoacrylate glue, with the connection covered by silver paint. The blocks were trimmed again, sputter-coated with a 40 nm layer of gold, and the top portion of the block was cut off for SEM imaging.

### Imaging and data analysis

Samples were imaged using a Thermo Fisher Scientific Volumescope™ SEM. Overview images were captured at 3.2 kV, 300 ns, with a pixel size of 12 nm (Supplementary Fig 1a, b) or 10 nm (Supplementary Fig. 1c, d). ROIs were imaged at 3.2 kV, using the VSDBS detector in low vacuum mode (30 Pa) with a 1 μs dwell time. ROIs from the ear tissue had a voxel size of $7 \times 7 \times 60$ nm, while those from the dorsal skin had a slice thickness of 80 nm. Image alignments and stitching were performed in FIJI TrakEM2[89]. Stitched datasets were imported into the Microscopy Image Browser for segmentation. Structures of interest were segmented manually using a combination of brush tools and AI-assisted segmentation via the Segment anything model 2, with super-pixel clustering used where appropriate to facilitate boundary detection[90,91]. The 3D models and surface measurements were generated and extracted using the AMIRA 3D software (version 2023 1.1, TFS). The contact sites' surfaces were visualized and measured as the surface intersection of the corresponding materials as follows: segmented models were imported into Amira software for 3D surface rendering and quantitative analysis. Surfaces were generated from the segmented volumes without additional binning or smoothing to preserve geometric accuracy. Contact interfaces between structures (e.g., *Borrelia*–pericyte, *Borrelia*–EC, pericyte–EC) were visualized using the Surface View module as intersections between two selected materials. While Amira by default renders surface intersections as contacts between materials and the exterior, we manually adjusted the selection to display true contact sites between specific structures. These interfaces were extracted using the Extract Surface tool and quantified with Amira's Surface Area Volume measurement tool.

### Reporting summary

Further information on research design is available in the Nature Portfolio Reporting Summary linked to this article.

## Data availability

The raw electron microscopy (SBF-SEM) data generated in this study have been deposited in the Figshare database under accession codes 30042805[92]; 30042550[93]; 30042646[94]; 30041983[95]; 30043204[96]; 30042739.v1[97]; 30043093[98]; 30042088[99]; 30042151[100]; 30042721[101].

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

## Acknowledgements

This work was supported by the Czech-BioImaging large RI project (LM2023050 and OP VVV CZ.02.1.01/0.0/0.0/18_046/0016045 funded by MEYS CR) (M.V.), the Czech Science Foundation Grant No. 22-18647K (M.V.), and the project reg. no. CZ.02.01.01/00/23_020/0008499, co-funded by the European Union (M.S.). The authors acknowledge Petra Masařová, Jiří Vaněček, and Helena Roháčková for their excellent technical support.

## Author contributions

M.V., M.S., and R.O.M.R. conceived and designed the project; M.V. and J.T. performed EM imaging; F.K. and J.K. conducted image analysis; M.S. and M.V. wrote the manuscript; R.O.M.R., J.T., and F.K. provided helpful suggestions and contributed to manuscript revision.

## Competing interests

The authors declare no competing interests.
