## [Transparent Peer Review file · Nature Communications]

Targeted volume imaging reveals early vascular interactions of Lyme disease pathogen in skin

Corresponding Author: Dr Marie Vancová

Version 1:

Reviewer comments:

Reviewer #1

(Remarks to the Author)

This manuscript describes a detailed morphologic reconstruction of skin and subcutaneous tissues from which *B. burgdorferi* added via needle inoculation after concurrent tick bite, gain access from the ECM into vessels that serve for dissemination. While much of the prior work on this extensively studied area has been done using morphologic approaches including in vivo imaging, in vitro models, and even 3D microvessels, the major advance in this work is the application of detailed SEM images reconstructed to better visualize how the spirochetes interact with the ECM, the vasculature, including both endothelial cells of capillaries and lymphatics, and also importantly pericytes. The imaging approach yields several remarkable observations that advance the understanding of how *B. burgdorferi* disseminates by careful measurements and 3 dimensional reconstructions of static images in great detail. The authors focus on 3 major areas: 1) interactions with pericytes that often precedes interactions with endothelial cells; 2) interactions with ECM components; and 3) interactions with immune cells extravasating from the vasculature. Perhaps the most striking observation is the detailed studies of how *B. burgdorferi* often interacts with pericytes, including penetration through and under the cells, presumably as a step toward more definitive interactions with endothelial cells. Surprisingly, many of the endothelial cells observed indicate that dissemination through lymphatics is a commonly occurring event. While no observations of *B. burgdorferi* penetrating through and into the lymphatic or vascular channels was observed, that this happens was noted with the presence of intact spirochetes in the lumens. The long delay engaged at the endothelial surface is similar to other in vitro model studies, and is suggest to be related to the difficulty of *B. burgdorferi* to transmigrate the basement membranes of the vessels. The authors also imply through relatively small numbers of examples, that *B. burgdorferi* transmigrates by both paracellular and transcellular mechanisms. Additionally, the studies beautifully demonstrate the migration of *B. burgdorferi* through collagenous bundles along their lengths that presumably minimizes the needed energy required for other routes, and events that demonstrate innate immune cell recruitment and engagement of *B. burgdorferi* within phagocytic events.

While these demonstrations are incredible in detail and provide support for several hypothetical mechanisms that could contribute to an approach that defines molecular and cellular events in the transition from ECM to disseminated tissue deposits, the evidence is solely morphological and does not specifically test hypotheses for inclusion or exclusion, resulting in a somewhat descriptive approach that is helpful, but still will require extensive alternative investigations to define the mechanisms of these observations. Overall, this is an outstanding descriptive work that can influence subsequent studies and an outstanding use of new ultrastructural tools to define \ subcellular host-pathogens interaction.

A few minor comments follow.

Page 7, line 228. Here the authors hypothesize that the tight spaces into which *B. burgdorferi* can squeeze might represent stationary adhesion to preclude phagocytosis by recruited innate immune cells. This seems a little speculative given that most innate immune cells have a ready capacity to squeeze through very tight spaces e.g. between endothelial cells with extravasation. Given this, why would this location provide any significant protection against the recruited innate immune cells?

Page 12, lines 382-383. Here, the author conjecture that given the role of pericytes in stabilizing endothelial barriers, that the insertion of *B. burgdorferi* between these two host cells, or entry through pericytes might compromise that cellular functions and barrier function. Perhaps this is so, but outside of the demonstrated barrier penetration by *B. burgdorferi* that the authors cite as evidence of barrier compromise, what other evidence is there that the bacterium compromises PC function and thus influences EC barriers further? Clearly there are enough examples of *Bb* interacting with PCs but not transmigrating? Is there any morphological evidence of altered PC function at these sites?

Page 12, lines 390-393. Here the authors speculate that the preference of pericytes for binding to fibronectin coupled with multiple *B. burgdorferi* fibronectin receptors could explain the propensity of interactions as suggested initially with pericytes

as a leverage toward subsequent endothelial cell interactions. However, given that these are static images that, despite the outstanding detail and resolution, still require some correlational observations, the observation still has limited evidence as support.

Reviewer #2

(Remarks to the Author)

This manuscript presents a compelling and technically innovative study using targeted, high-resolution serial block-face scanning electron microscopy (SBF-SEM) to visualize early *Borrelia burgdorferi* (Bb) interactions in host tissue at the tick bite site. The use of this imaging modality is a significant advance for the field of Lyme disease research. Demonstrating spatial context to rare but biologically important events during early infection address a gap in our understanding of how the bacterium begins its dissemination.

Major Strengths:

The observations of Bb aligning with endothelial cells (ECs) rather than forcefully breaching them, along with the demonstration of both transcellular and paracellular migration across lymphatic vessels, extend the current models of dissemination. The interaction with pericytes and preference for lymphatic rather than blood vessel intravasation are particularly interesting and underexplored areas.

The application of SBF-SEM in this context is novel and powerful. It allows for 3D reconstructions of host-pathogen interactions at cellular and subcellular resolution, which has rarely been achieved in this field. This technique complements and corroborates previous findings from other imaging and molecular approaches.

The manuscript also thoughtfully builds on and confirms prior studies, including Bb internalization by host cells and its use of both trans- and paracellular routes.

Points for Revision:

1. Clarification of Cell Type Identification in EM: The identification of pericytes, endothelial cells, and immune cell types (e.g., neutrophils, mast cells) in electron micrographs could benefit from clearer criteria. Consider referencing established ultrastructural features used for such classifications in skin/tissue EM. For readers less familiar with EM, additional labeling in representative images or a supplementary guide would enhance accessibility.
2. Differentiation Between Vascular Structures: Please provide more details or citations regarding how lymphatic vessels were distinguished from blood vessels in the EM data. The text refers to differences in basement membrane organization and continuity, but elaborating on the diagnostic morphological traits would strengthen the interpretations.
3. Details on Image/Data Analysis: The methodology section could be expanded to explain how imaging data were processed, annotated, and quantified within the materials and methods. Some of this is within the results, e.g., line 186-180, but the materials and methods seems light. What does "The contact sites surfaces were visualized and measured as the surface intersection of the corresponding materials" mean?
4. Citation Suggestions: Line 333–334: Casselli et al., 2021 also reported motion reversal events in *B. burgdorferi* along collagen in the dura mater.

Line 350–354: The idea that *Borrelia* remodels its surface in the tick prior to transmission is intriguing and is supported by the known upregulation of surface lipoproteins during the blood meal.

The discussion of collagen association: *B. burgdorferi* has a known collagen-binding protein which is required for infectivity in the mouse model (e.g., Zhi et al., 2015).

Minor Comments:

Line 66- can the authors clarify what they mean here?

The finding that outer membrane vesicle (OMV) shedding was limited is interesting and might merit further discussion — is there any other data regarding OMV shedding in the mouse?

Conclusion:

This is a well-written manuscript that will be of broad interest to researchers in the Lyme disease field. Some clarifications are needed, particularly regarding image interpretation and vessel classification.

Best regards,
Catherine Brissette

Reviewer #3

(Remarks to the Author)

In their study, "Targeted volume imaging reveals early vascular interactions of Lyme disease pathogen in skin," the authors present vEM data and surface renderings of *Borrelia* spirochetes in the complex environment of "real tissue" around tick bites. In blocks with several hundred μm edge length, they show examples of various stages of parasite-host interaction in high 3D resolution: *Borrelia* bypasses rigid obstacles, contacts pericytes in an alongside alignment, or slips into gaps between cell layers and is taken up by neutrophils. The morphological data are discussed in terms of an infection mechanism based on parasite behavior that avoids mechanical obstacles and displays little (if any) enzymatic activity.

The study provides new and very interesting results on a parasitologically highly relevant topic (and is particularly exciting for a blockface electron microscopist who was recently plagued by symptoms of Lyme disease himself) based on a relatively

new method that is perfectly suited to study spirochetes in the 3D context of the affected tissue.

The EM images and renderings are of very good quality due to extensive and precise work (e.g., grueling pericyte segmentation). Only Fig. 3d appears to exhibit slight astigmatism, which, however, is acceptable in the context of all other high-quality images.

All together it is a real asset for the professional community to have visualizations of this quality to better understand invasion and spread of spirochaetes within the host. I recommend publication in Nature Communications as soon as a number of minor corrections and clarifications have been made:

Abstract and Summary are somehow similar.

Are both sections required separately for the journal you are applying for?

Line 30: Never use unexplained abbreviations in the summary! Spell out ECM (see line 74) and OMV (see line 103) at first use.

Lines 45, 89, 305: The undisputed advantage of the present study over other in vitro studies is the investigation of parasite penetration behavior in a true histological context, i.e. in vivo during the first steps of sample preparation. The 3DEM examination (slicing + imaging) is naturally not carried out in vivo but on dead material and shows a snapshot of the living event at the time of fixation. So you have to be very careful when you can use the term in vivo and when not. It is true that you get high-resolution insights of host-pathogen interactions in vivo ... the investigation of tissue using 3D EM imaging, however, is NOT in vivo. "Dissemination in vivo" in line 264 describes a process before fixation and is therefore correct in this context.

Line 64: contentious \Leftrightarrow but (logic of the sentence?)

Line 86-88: this sentence needs citations (more than one or a review article)

Line 118: What volume did you inject? How exactly did you determine the number of Bb cells in your injections fluid?

Line 126: how long fixed? at RT?

Line 139: interesting: did you jump from 100% EtOH to 30% acetone in water or 30% acetone in EtOH?

Line 146: gold: how thick?

Line 148/165:

1. How many pixels (x + y) did you record in your overview images for both samples ... or did you stitch several overlapping tiles? For 1000 μm x 450 μm @ 12 nm resolution you would need a single image of 83 333 x 37 500 pixels. The ApreoVS that I use allows a maximum of 40 960 x 40 960 px but the distortion and defocussing would be severe at the image margins. Vignetting in Suppl. Fig 1a shows somehow irregularly arranged tiles in the left part of the image, Suppl. Fig 1c shows tiles of ca. 91 x 91 μm with the given scale bar of 200 μm (this would correspond to something like 8192 x 8192 px @ 12 nm/px with 4% overlap?!)

2. How many pixels (x + y) did you record for the detailed ROIs @ 7 nm resolution?

Line 156: Amira version number?

Line 157: Which Amira tool did you use to precisely quantify surface-contact-areas in 3D?

Lines 169 – 180: The interactive approach to find small relevant ROIs in a bigger volume is smart ... although a common part of the user training at the ApreoVS (did you use MAPS software?). Again, here it would be interesting how many substacks you recorded and how many planes (from – to).

Lines 441/442: Any definitive statement on the subject of "moving or not moving" is speculative for fixed tissue. Better write "it is likely that ..."

Line 480: Just to know: is there anything known about the role of tick saliva for the microenvironment of the bite area?

Line 732: Bracket) missing

Figure 2+3, Suppl Fig. 2-4, Suppl Movie 1-3: You write "interacting" (= interpretation), but in fact you see "contacting" (= pure description). Being close to something does not necessarily mean that it interacts. In the case of invaginating Bb-pericyte interactions, I completely agree with the authors; in the other cases, I would choose a more cautious formulation. The same applies to the "Results" chapter, lines 208 ff.

However, a convincing explanation of the interaction for a specific case of contact is given in lines 227/228.

Line 748: How can you know that the Bb cell is stationary – are contacts fixed or transient? With a single fixed sample there is no temporal resolution.

Line 749: in dorsal

Figure 4: If the black arrow in Fig. 4c indicates the unidentified structure, please cite it in line 761.

Line 775: OMV (black arrows) are depicted in Fig 5a, but described for Fig. 5b

Line 795: There is only 1 black arrow.

In the overview Fig. 7b I cannot clearly see a penetration of the BL by the neutrophile cell, at least not in this single plane.

Line 803: Black arrows indicate tapering protrusions between nuclear segments (here you have chromatin PLUS nuclear membrane and not only "chromatine strands")

Line 804: Use . , ; properly

Lines 812, 813: For your mechanism is hypothetic this maybe should be stated with one word. E.g. "probing" ist probable, but you cannot prove it with your purey morphological data.

Line 842: 1x ; instead of , ... 1x , missing

Version 2:

Reviewer comments:

Reviewer #2

(Remarks to the Author)

All of my concerns have been addressed

Reviewer #3

(Remarks to the Author)

All my concerns have been addressed. Well done!

Reviewer #1:

We appreciate the reviewer's positive assessment of our morphological approach. We agree that future studies combining high-resolution imaging with functional and molecular assays will be necessary to fully disentangle the molecular mechanisms suggested by our observations.

Page 7, line 228. Here the authors hypothesize that the tight spaces into which *B. burgdorferi* can squeeze might represent stationary adhesion to preclude phagocytosis by recruited innate immune cells. This seems a little speculative given that most innate immune cells have a ready capacity to squeeze through very tight spaces e.g. between endothelial cells with extravasation. Given this, why would this location provide any significant protection against the recruited innate immune cells?

We recognize that this remains a hypothesis and have explicitly stated it as such. Our reasoning is based on the following considerations:

It is well established that innate immune cells (IICs) can squeeze through extremely confined spaces within tissues/during extravasation. What remains less clear, however, is whether they can perform phagocytosis during these "squeezed state" transigrations. Effective phagocytosis by macrophages and other professional phagocytes requires the formation of large, actin-driven membrane protrusions that wrap around the target. When a macrophage is physically constrained in a tight space, its ability to extend these protrusions may be hindered, potentially impairing its phagocytic function.

Also, in the absence of specific chemotactic signals, IICs patrol tissues in a random, exploratory manner, continuously scanning for apoptotic cells, pathogens, and other danger-associated signals. However, IICs choose the path of least resistance when moving through confined space (PMID: 35810162). Therefore, pathogens that remain hidden in confined spaces (and not producing any chemotactic signals, eg. outer membrane vesicles)—and avoid exposing themselves to random contact with these immune cells—may increase their chances of evading detection and clearance.

Page 12, lines 382-383. Here, the author conjecture that given the role of pericytes in stabilizing endothelial barriers, that the insertion of *B. burgdorferi* between these two host cells, or entry through pericytes might compromise that cellular functions and barrier function. Perhaps this is so, but outside of the demonstrated barrier penetration by *B. burgdorferi* that the authors cite as evidence of barrier compromise, what other evidence is there that the bacterium compromises PC function and thus influences EC barriers further? Clearly there are enough examples

of Bb interacting with PCs but not transmigrating? Is there any morphological evidence of altered PC function at these sites?

Pericytes comprise a very heterogeneous population of cells (PMC10936204). During image analysis, we searched for morphological signs of barrier disruption or altered pericyte function. However, the natural variability in the pericyte–endothelial cell interface made interpretation challenging. A recent study shows that pathogenic bacteria can modulate the expression profile of endothelial cells through interactions with pericytes, thereby dampening endothelial pro-inflammatory responses (Carvalho et al., 2024; PMC10936204). Thus, despite the absence of clear morphological signs, functional disturbances are likely still occurring. In the discussion, we have carefully worded our interpretation to indicate that barrier destabilization is a possible outcome, though not a definitive conclusion.

Page 12, lines 390-393. Here the authors speculate that the preference of pericytes for binding to fibronectin coupled with multiple B. burgdorferi fibronectin receptors could explain the propensity of interactions as suggested initially with pericytes as a leverage toward subsequent endothelial cell interactions. However, given that these are static images that, despite the outstanding detail and resolution, still require some correlational observations, the observation still has limited evidence as support.

We fully agree that further experiments are necessary to identify the key molecules mediating the interaction between pericytes and Bb. We plan for future experiments to use cultured pericytes to investigate potential molecular drivers of this interplay based on the results obtained here. We believe that the way we phrased the discussion, along with the references cited, makes it clear that the involvement of fibronectin in the binding process is a plausible possibility, but not definitively established.

Reviewer #2

We sincerely value the positive appraisal our study has received.

1. Clarification of Cell Type Identification in EM: The identification of pericytes, endothelial cells, and immune cell types (e.g., neutrophils, mast cells) in electron micrographs could benefit from clearer criteria. Consider referencing established ultrastructural features used for such classifications in skin/tissue EM. For readers less familiar with EM, additional labeling in representative images or a supplementary guide would enhance accessibility.

We have revised Supplementary Figure 1 to include additional images e-i showing representative examples of key cell types (pericytes, endothelial cells, mast cells, neutrophils, and fibroblasts), with labelled characteristic ultrastructural features used for their identification. We have also updated the figure legend accordingly and added a sentence for clarification to the Results section (**line 185-187**): “Characteristic features used for the identification of key cell types are shown in Supplementary Fig. 1e-i. A more detailed description of the morphology of blood and lymphatic capillaries is provided in previous studies^{ref31,32}”

Updated legend to Suppl Fig 1”..... **e, f** Representative image of a lymphatic capillary (LC) and a blood capillary (BC). The BC typically contains erythrocytes (ERY) and other blood cells within its lumen (L) and is lined by endothelial cells (EC) that form electron-dense tight junctions (gray arrow) between neighbouring cells and contain numerous endocytic vesicles (black arrows) in their cytoplasm. The apical surface of ECs is covered by a basement membrane (BM), which is shared with a pericyte (PC). The PC extends long, thin processes (blue arrowheads) that wrap around the BM and may invaginate into the EC (white arrow). The LC is lined by flat lymphatic endothelial cells (LEC) that form outward protrusions (see gray arrows in Fig. 5a). The LC lumen typically appears empty or may contain non-erythrocytic cells. The BM is barely discernible. **g** Representative image of a mast cell with a monolobed nucleus (N), filopodia (black arrows), and cytoplasm densely packed with vesicles (v) of variable electron density and diameter. **h** Representative image of a neutrophil with a multilobed nucleus (N), phagolysosomes (white arrow), filopodia (black arrow), and finely granular, electron-dense cytoplasm; lamellipodia are not visible in this section. **i** Fibroblast with an oval nucleus (N) and, in this section, two long, thin cytoplasmic processes (black arrows), which may be branched or elaborately extended. e-i Ear.”

2. Differentiation Between Vascular Structures: Please provide more details or citations regarding how lymphatic vessels were distinguished from blood vessels in the EM data. The text refers to differences in basement membrane organization and

continuity, but elaborating on the diagnostic morphological traits would strengthen the interpretations.

We have now included two references (PMID: 33753399, PMID: 35534209) that provide a detailed description of the morphology of both types of vessels. Please see our response to the previous comment.

3. Details on Image/Data Analysis: The methodology section could be expanded to explain how imaging data were processed, annotated, and quantified within the materials and methods. Some of this is within the results, e.g., line 186-180, but the materials and methods seems light. What does “The contact sites surfaces were visualized and measured as the surface intersection of the corresponding materials” mean?

Thank you for the comment, also based on suggestion of reviewer 3, we have added several sentences to the Imaging and Data Analysis section of the Methods: “Stitched datasets were imported into the Microscopy Image Browser (MIB) for segmentation. Structures of interest were segmented manually using a combination of brush tools and AI assisted segmentation via the Segment anything model 2, with super-pixel clustering used where appropriate to facilitate boundary detection^{27,28}. The 3D models and surface measurements were generated and extracted using the AMIRA software (version 2023 1.1, TFS). The contact sites surfaces were visualized and measured as the surface intersection of the corresponding materials as follows: segmented models were imported into Amira software for 3D surface rendering and quantitative analysis. Surfaces were generated from the segmented volumes without additional binning or smoothing to preserve geometric accuracy. Contact interfaces between structures (e.g., *Borrelia*–pericyte, *Borrelia*–endothelial cell, pericyte–endothelial cell) were visualized using the Surface View module as intersections between two selected materials. While Amira by default renders surface intersections as contacts between materials and the exterior, we manually adjusted the selection to display true contact sites between specific structures. These interfaces were extracted using the Extract Surface tool and quantified with Amira’s Surface Area Volume measurement tool.” (line 141-156).

4. Citation Suggestions: Line 333–334: Casselli et al., 2021 also reported motion reversal events in *B. burgdorferi* along collagen in the dura mater.

The reference has been added (line 339, ref 46).

Line 350–354: The idea that *Borrelia* remodels its surface in the tick prior to transmission is intriguing and is supported by the known upregulation of surface lipoproteins during the blood meal.

Thank you.

The discussion of collagen association: B. burgdorferi has a known collagen-binding protein which is required for infectivity in the mouse model (e.g., Zhi et al., 2015).

The importance of collagen-binding activity of Bb for mouse infection has been highlighted and the reference added **(line 333-334, ref 45)**.

Minor Comments:

Line 66- can the authors clarify what they mean here?

We appreciate the comment and would like to clarify: HUVECs orient their basolateral side toward the structural support (basement membrane/ECM). The same happens in vitro e.g. in Transwell inserts, where HUVECs orient their basolateral side toward the porous membrane. By adding Bb on the apical side of HUVECs in these systems, we then basically mimic the extravasation process.

Therefore, for greater clarity, we have extended the sentence: “Notably, all these studies were modelled in a way that they follow the process of extravasation - introducing Bb on the apical side of cultured ECs or into the lumen of the vessels and tracking its movement toward the basal side of ECs/perivascular space.” **(line 45-46)**.

The finding that outer membrane vesicle (OMV) shedding was limited is interesting and might merit further discussion — is there any other data regarding OMV shedding in the mouse?

Indeed, it would be very interesting to discuss more this topic, but unfortunately, we are unaware of any data in the literature on the OMV shedding in mice. While Bb is well-documented to produce OMVs both in vitro and during tick engorgement, direct evidence of OMV shedding in mice appears lacking.

Reviewer #3:

We sincerely appreciate the kind appraisal of our work and the endorsement for publication in *Nature Communications*.

Abstract and Summary are somehow similar.

Are both sections required separately for the journal you are applying for?

Following the Nature Communications formatting guidelines, we integrated the former "Summary" section into the Discussion (**lines 499-517**).

Line 30: Never use unexplained abbreviations in the summary! Spell out ECM (see line 74) and OMV (see line 103) at first use.

Thank you for noticing. The Summary section is no longer in the manuscript, as mentioned above.

Lines 45, 89, 305: The undisputed advantage of the present study over other in vitro studies is the investigation of parasite penetration behavior in a true histological context, i.e. in vivo during the first steps of sample preparation. The 3DEM examination (slicing + imaging) is naturally not carried out in vivo but on dead material and shows a snapshot of the living event at the time of fixation. So you have to be very careful when you can use the term in vivo and when not. It is true that you get high-resolution insights of host-pathogen interactions in vivo ... the investigation of tissue using 3D EM imaging, however, is NOT in vivo.

“Dissemination in vivo” in line 264 describes a process before fixation and is therefore correct in this context.

Line 45: By studying critical host-pathogen interactions in 3D at high resolution in vivo, we can gain a much deeper understanding of the mechanisms driving pathogenesis.

Replaced by: “By studying critical host-pathogen interactions in 3D at high resolution within the native tissue context, we can gain a much deeper understanding of the mechanisms driving pathogenesis.” (**line 516-517**)

Line 89: Despite its growing popularity, high-resolution, large-volume structural insights into pathogen-host interactions in vivo remain largely unexplored

Replaced by: “Despite its growing popularity, high-resolution, large-volume structural insights into pathogen interactions within their native host environment remain largely unexplored.” (**line 69**)

Line 305: To define the spatio-temporal features of microbial infections, structural data need to be obtained in vivo in three dimensions with high resolution.

Replaced by: “To define the spatio-temporal features of microbial infections, structural data need to be obtained in the natural host environment in three dimensions with high resolution.” **(line 306-307)**

Line 64: contentious <=> but (logic of the sentence?)

Thank you for the correction. New sentence is: “The literature is contentious, and it remains unclear whether vascular transmigration occurs primarily via transcellular or paracellular pathways”. **(line 42)**

Line 86-88: this sentence needs citations (more than one or a review article)

We have included the citation: Peddie CJ, Genoud C, Kreshuk A, Meechan K, Micheva KD, Narayan K, Pape C, Parton RG, Schieber NL, Schwab Y, Titze B, Verkade P, Aubrey A, Collinson LM. Volume electron microscopy. Nat Rev Methods Primers. 2022 Jul 7;2:51. doi: 10.1038/s43586-022-00131-9. **(line 68, ref 23)**

Line 118: What volume did you inject? How exactly did you determine the number of Bb cells in your injections fluid?

“The bacteria were enumerated using a Petroff-Hausser counting chamber, and the injection volume was adjusted to 50 µL.” This sentence has been added to the M&M section. **(line 103)**

Line 126: how long fixed? at RT?

The sentence has been reworded as follows: “These samples were immediately fixed in 2.5 % glutaraldehyde, 2 % formaldehyde, and 2 mM calcium chloride in 0.15 M cacodylate buffer for 2 hours at room temperature, followed by overnight fixation at 4 °C.” **(line 113-114)**

Line 139: interesting: did you jump from 100% EtOH to 30% acetone in water or 30% acetone in EtOH?

Thank you for this comment. We apologize for the mistake. The samples were dehydrated only in acetone. The sentence has been now corrected as follows: “The samples were then dehydrated in a graded acetone series (30 %, 50 %, 70 %, 90 %, 100 %) for 2 × 7 minutes at 4°C at each step. The repetitive part: “ ..followed by a 30–100 % acetone series” has been removed. **(line 125)**

Line 146: gold: how thick?

The missing information has been added. The sentence has been reworded as follows: “The blocks were trimmed again, sputter-coated with a 40 nm layer of gold, and the top

portion of the block was cut off for SEM imaging.” (line 133)

Line 148/165:

1. How many pixels (x + y) did you record in your overview images for both samples ... or did you stitch several overlapping tiles? For 1000 μm x 450 μm @ 12 nm resolution you would need a single image of 83 333 x 37 500 pixels. The ApreoVS that I use allows a maximum of 40 960 x 40 960 px but the distortion and defocussing would be severe at the image margins.

Vignetting in Suppl. Fig 1a shows somehow irregularly arranged tiles in the left part of the image, Suppl. Fig 1c shows tiles of ca. 91 x 91 μm with the given scale bar of 200 μm (this would correspond to something like 8192 x 8192 px @ 12 nm/px with 4% overlap?!)

The overview images were generated by stitching tiles of 10,240 \times 10,240 pixels with 20% overlap. The irregular stitching in the right part of Supplementary Fig. 1a is caused by a combination of charging on pure resin and the absence of structural landmarks. As a result, both tile registration during acquisition and post-acquisition stitching were less reliable in these areas. Importantly, the regions containing tissue were well stitched and unaffected by these artifacts.

We also re-checked the acquisition settings for the overview in Supplementary Fig. 1c, d and corrected the previously stated pixel size from 12 nm to 10 nm. We apologize for this mistake. The sentence in lines 133–134 has been updated accordingly:

“Overview images were captured at 3.2 kV, 300 ns, with a pixel size of 12 nm (Supplementary Fig. 1a, b) and 10 nm (Supplementary Fig. 1c, d).” (lines 137-138)

2. How many pixels (x + y) did you record for the detailed ROIs @ 7 nm resolution?

Detailed ROIs were acquired at a constant resolution of 7 nm, with pixel dimensions varying between 8192 \times 8192, 6144 \times 6144, 4096 \times 4096, and 6144 \times 4096.

Line 156: Amira version number?

The missing information has been added. The sentence has been reworded as follows: “The 3D models and surface measurements were generated and extracted using the AMIRA software (version 2023 1.1, TFS).” (line 146)

Line 157: Which Amira tool did you use to precisely quantify surface-contact-areas in 3D?

We have added the information to the text in the materials and methods: section “The contact sites surfaces were visualized and measured as the surface intersection of the corresponding materials as follows: segmented models were imported into Amira

software for 3D surface rendering and quantitative analysis. To preserve geometric accuracy, surfaces were generated from the segmented volumes without additional binning or smoothing. Contact interfaces between structures (e.g., Borrelia–pericyte, Borrelia–endothelial cell, pericyte–endothelial cell) were visualized using the Surface View module as intersections between two selected materials. While Amira by default renders surface intersections as contacts between materials and the exterior, we manually adjusted the selection to display true contact sites between specific structures. These interfaces were extracted using the Extract Surface tool and quantified with Amira’s Surface Area Volume measurement tool.” (lines 148-158)

Lines 169 – 180: The interactive approach to find small relevant ROIs in a bigger volume is smart ... although a common part of the user training at the ApreoVS (did you use MAPS software?). Again, here it would be interesting how many substacks you recorded and how many planes (from – to).

Yes, we used the MAPS software. The reason why we highlight this approach is to explain it to those who are not familiar with working at the Apreo VS and MAPS software, and this approach was extremely efficient in this case thanks to the fact that the Borrelia are elongated. We estimate that we recorded more than 300 substacks in total, and the number of planes/sections within individual ROIs differed (depending on whether we hit the end or the beginning of a Borrelia, and how interesting the region was).

Lines 441/442: Any definitive statement on the subject of “moving or not moving” is speculative for fixed tissue. Better write “it is likely that ...”

The text has been revised according to the suggestion.

Line 480: Just to know: is there anything known about the role of tick saliva for the microenvironment of the bite area?

The saliva of ticks contains numerous bioactive compounds that enable successful feeding and pathogen transmission by reshaping the host’s local environment. These molecules can suppress immune responses, prevent blood clotting, and alter vascular integrity. Emerging evidence also points to their involvement in disrupting ECM dynamics and modulating endothelial permeability (see: <https://doi.org/10.3389/fimmu.2025.1520665>). While this is an important factor in Borrelia dissemination, it lies outside the scope of our current study and is therefore not further discussed.

Line 732: Bracket) missing

Corrected. Thank you.

Figure 2+3, Suppl Fig. 2-4, Suppl Movie 1-3: You write “interacting” (= interpretation), but in fact you see “contacting” (= pure description). Being close to something does not necessarily mean that it interacts. In the case of invaginating Bb-pericyte interactions, I completely agree with the authors; in the other cases, I would choose a more cautious formulation. The same applies to the "Results" chapter, lines 208 ff.

However, a convincing explanation of the interaction for a specific case of contact is given in lines 227/228.

Thank you for pointing out these important details. The suggested revisions have been implemented in the figure legends and main text (**line 210-212**).

The legends have been corrected as follow:

Fig. 2: Pericyte targeting by *B. burgdorferi*. Representative images from SBF-SEM datasets are shown in grayscale, with corresponding 3D models. The percentages indicate the Bb surface areas in contact with the pericyte (blue arrows) and endothelial cells (green arrows).

Fig. 3: Multisite adhesion of *B. burgdorferi* to the blood capillary surface in dorsal skin.

Supplementary Fig. 2: *B. burgdorferi* in contact with the pericyte projections in the dorsal skin.

Supplementary Fig. 3: Intimate contacts of *B. burgdorferi* with the blood capillary surface in dorsal skin.

Supplementary Fig. 4: *B. burgdorferi* in contact with the blood capillary surface in dorsal skin.

Supplementary Movie 1 SBF-SEM dataset showing *B. burgdorferi* in contact with a pericyte projection. For further details, see Fig. 2a.

Supplementary Movie 2 SBF-SEM dataset shows *B. burgdorferi* in contact with a pericyte projection. For further details, see Fig. 2b.

Supplementary Movie 3 SBF-SEM dataset demonstrates *B. burgdorferi* in contact with a blood capillary. For further details, see Fig. 4.

Line 748: How can you know that the Bb cell is stationary – are contacts fixed or transient? With a single fixed sample there is no temporal resolution.

Agreed. Based on the localization of *Borrelia* and the number of contact points with endothelial cells, we interpreted this as stationary adhesion. However, since the analysis was performed on fixed tissue samples, we cannot confirm this with certainty. The legend to Fig. 3 has been revised to: Multisite adhesion of *B. burgdorferi* to the blood capillary surface in dorsal skin.

Line 749: in dorsal

Corrected.

Figure 4: If the black arrow in Fig. 4c indicates the unidentified structure, please cite it in line 761.

The legend has been corrected: “. c Representative slices from the region of dataset indicated by black arrowheads in b reveal an unidentified structure adjacent to the *Bb* (black arrow).” (line 816)

Line 775: OMV (black arrows) are depicted in Fig 5a, but described for Fig. 5b

The legend has been corrected (line 829).

Line 795: There is only 1 black arrow. In the overview Fig. 7b I cannot clearly see a penetration of the BL by the neutrophil cell, at least not in this single plane.

Thank you for the comment. We have added a second black arrow to Fig 7b and included an insert enlarged part in the next plane of the image to better demonstrate the penetration of the immune cell through the basement membrane of the blood vessel. The figure legend has been updated: “Black arrows indicate areas where neutrophils traverse the basement membrane, which represents the narrowest area through which the immune cells pass. In the inset, this region is shown in a different plane. For further details, see supplementary movie 15.” (Line 849)

Line 803: Black arrows indicate tapering protrusions between nuclear segments (here you have chromatin PLUS nuclear membrane and not only “chromatine strands”)

Thank you for the comment. The sentence has been reworded: “Black arrows indicate tapering protrusions between nuclear segments”. (line 856)

Line 804: Use . , ; properly

Corrected.

Lines 812, 813: For your mechanism is hypothetic this maybe should be stated with one word. E.g. “probing” ist probable, but you cannot prove it with your purey morphological data.

Thank you for the comment. We agree that our interpretation is based solely on morphological data and have therefore adjusted the title and legend of Fig 8 to avoid overinterpretation.

Line 842: 1x ; instead of , ... 1x , missing

Corrected. Thank you.